# Inclusive Healthcare System for Children with Disabilities: A Bibliometric Analysis and Visualization

**DOI:** 10.3390/healthcare13172106

**Published:** 2025-08-24

**Authors:** Erkan Gulgosteren, Yavuz Onturk, Abdullah Cuhadar, Mihaela Zahiu, Monica Stanescu, Rares Stanescu

**Affiliations:** 1Faculty of Sports Sciences, Mersin University, Mersin 33010, Türkiye; egulgosteren@mersin.edu.tr; 2Faculty of Sports Sciences, Yalova University, Yalova 77400, Türkiye; yavuzonturk@hotmail.com; 3Faculty of Applied Sciences, Karamanoğlu Mehmetbey University, Karaman 70100, Türkiye; abdullahcuhadar@kmu.edu.tr; 4Faculty of Physical Education and Sports, National University of Physical Education and Sports, 060057 Bucharest, Romania; mihaela.zahiu@gmail.com (M.Z.); raresuniv@yahoo.com (R.S.); 5Doctoral School, National University of Physical Education and Sports, 060057 Bucharest, Romania

**Keywords:** children with disabilities, healthcare services, bibliometric analysis, special laws

## Abstract

Background: Children with disabilities face complex, systemic health access barriers rooted in societal, institutional, and structural inequities, requiring urgent global policy attention. Publications on access to health services for this population category have been found to have a significant growth in both quantity and content. The article aims to examine the structure and evolution of scientific literature in analyzing the healthcare system through the lens of inclusive services. Methods: We present the bibliometric profile of the global literature on access to health services for children with disabilities, the publication trends, the structure of research in this field concerning geographical distribution, methodological approaches, and interdisciplinary collaborations, and the core research topics, conceptual clusters, and future research directions in the field. The publications were screened from Web of Science databases, using PRISMA methodology. Finally, 1100 academic publications published between 1984 and 2025, obtained from a total of 432 different sources, the majority of which were peer-reviewed journals, were screened. Results: The calculated annual publication growth rate of 8.37% and the distinct upward trend observed, especially after 2015. The highest level was reached in 2023, with over 90 publications showing that the topic has become a focus of international academic interest. The USA (33.5%), the United Kingdom (15.7%), Australia (9.5%), and Canada (9.5%) stood out in publications, and there were strong collaborative networks among European nations (8.2%). Conclusions: Although high-income countries still appear to play a dominant role in research production, expanding international collaborations and distributing resources more equitably will contribute to the development of more inclusive solutions on a global scale. Temporal trends show an evolution toward diagnostic processes, family-centered approaches, and psychosocial dimensions. The results draw a clear picture of the current research landscape regarding access to health services for pediatric disability populations and identify potential directions for future research.

## 1. Introduction

A significant and long-debated issue in contemporary health policy is the challenge children with disabilities face in accessing health services. Data from the World Health Organization’s 2022 “Global Report on Health Equity for Persons with Disabilities” confirms that 1.3 billion people globally experience significant disability, a figure corresponding to 16% of the world’s population [1]. Children constitute a substantial portion of this number. As stated in UNICEF’s 2021 “Seen, Counted, Included” report, approximately 240 million children live with various forms of disability [2]. Such figures indicate that one in every ten children has a disability, which renders the elimination of health service inequities for this group a key requisite for both fundamental human rights and sustainable development goals. A comprehensive global scoping study demonstrates the multidimensional nature of systemic barriers that individuals with disabilities encounter in accessing healthcare [3] and another one, emphasized, disability during childhood and adolescence stands out as an area that requires greater priority on the global health agenda [4].

The health injustices experienced by pediatric populations with disabilities are too complex to be explained by medical reasons alone, and that complexity is highlighted in the WHO’s global report, which documents that individuals with disabilities often face premature death risks of up to 20 years, experience significantly poorer health outcomes, and encounter substantial limitations in daily functioning [1]. Epidemiological evidence shows that children with disabilities have a two-fold higher risk of developing chronic conditions such as asthma, depression, diabetes, obesity, oral diseases, and stroke as compared to their non-disabled peers. UNICEF’s analysis corroborates this as it reports that children facing disability-related challenges are 53% more likely to present with acute respiratory tract infection symptoms, 51% more likely to experience unhappiness, and 41% more likely to feel discriminated against [2].

Root causes of those health inequities are not confined to the disability itself but stem from multidimensional factors such as societal prejudice, institutional discrimination, poverty, exclusion from education and employment, and structural problems within the health system. The Levesque’s conceptual framework for access to health services offers a valuable theoretical perspective for analyzing this multilayered phenomenon—it considers access to healthcare as a patient-centered process and examines the interaction between five dimensions of accessibility (approachability, acceptability, availability-accommodation, affordability, and appropriateness) and five corresponding population capacities (ability to perceive, seek, reach, pay, and engage) [5]. Applying this theoretical framework specifically to minors with disabilities allows for a thorough analysis of access barriers by integrating both health system and patient perspectives.

Recent large-scale research for the WHO Global Report categorized barriers to healthcare access for people with disabilities under six broad health system components: (1) health and care workforce, (2) health information systems, (3) health system financing, (4) leadership and governance, (5) service delivery, and (6) essential medicines and equipment [1]. This methodical work covers research from 2011 to 2022 and recommendations from WHO consultation meetings involving more than 1250 experts [1]. The study’s exhaustive scope discloses the multilayered nature of structural barriers in healthcare access for children with complex health needs.

The original contributions of our study are focused on three main areas. Firstly, from an academic perspective, we systematically identify knowledge gaps, determine research priorities, and define opportunities for interdisciplinary collaboration. Secondly, from a policy development standpoint, we provide an objective assessment of the current situation regarding healthcare access for children with disabilities and contribute to evidence-based policymaking processes. Thirdly, from a methodological viewpoint, our study is the first comprehensive bibliometric analysis conducted in the field of health services for children in this demographic, and it establishes a methodological foundation for future research.

Analyses of the literature show that half (50.6%) of studies on access to healthcare services for individuals with disabilities use qualitative methods, a quarter (25.3%) use mixed methods, 15.7% use a quantitative approach, and 8.4% are systematic reviews. Most studies examined mixed disability types (33.7%), followed by psychosocial (25.3%), physical (16.9%), intellectual (14.5%), and sensory disabilities (9.6%). Nearly half of the studies (54.2%) provide information on specific groups at higher risk of exclusion, with a particular focus on women (19.3%) and children with disabilities (13.3%).

From a geographical perspective, research on healthcare access for individuals with disabilities is largely concentrated in the Americas (28.9%) and the Europe-Western Pacific (19.3%) regions. In the case of Türkiye, the Eastern Mediterranean region was found to be the least studied area (3.6%). It is noteworthy that 66.3% of the studies were carried out in high-income countries, whereas only 3.6% were performed in low-income countries. These findings indicate a substantial research gap in terms of global health equity.

In Türkiye, access to health services for pediatric populations who have disabilities is based on a broad legal and institutional framework coordinated by the Ministry of Family and Social Services. The “Special Needs Assessment Report for Children” (ÇÖZGER) system functions as a national evaluation mechanism to identify the special needs of children aged 0–18 (Ministry of Family and Social Services, 2023) [6]. Within this system, children are assessed in categories such as “Mild Special Needs,” “Significant Special Needs,” and “Very Profound Special Needs” to determine their rights to education, health, and social services. Individuals with disabilities also have priority status when receiving healthcare in Türkiye. Citizens registered as “priority” with the Ministry of Health can book appointments through special quotas.

However, a 2018 report by UNICEF and the European Disability Forum covering Türkiye points to significant systemic problems, stating that children with disabilities are “excluded from birth” and experience inadequate access to early diagnosis and intervention programs, community support, health services, and genuinely inclusive education [7]. Similar concerns were noted in Türkiye’s evaluation by the UN Committee on the Rights of Persons with Disabilities. Although progress in inclusive education policies was acknowledged, the fact that over 30% of children within the disability community do not participate in any level of education is seen as a serious problem [8].

In Romania, the access to healthcare services for children with disabilities is clearly regulated by Law no. 448/2006, Law no. 95/2006, and Law no. 272/2004, and reinforced by the UN Convention on the Rights of Persons with Disabilities [9]. These laws guarantee free, prioritized, and adapted access to medical care and rehabilitation. They impose clear obligations on authorities to ensure nondiscriminatory access, tailored support, and integration into the community. In practice, the authorities are obliged to provide not only treatment but also prevention, early diagnosis, and ongoing support services.

Law no. 448/2006 on the Protection and Promotion of the Rights of Persons with Disabilities is the main law that classifies disabilities and establishes the corresponding rights: children with mild disabilities, children with moderate disabilities, children with severe disabilities, and children with pronounced disabilities. The classification is determined following a medical-psycho-social assessment, carried out by the Child Protection Commission, based on a certificate of disability classification. Also, the inclusion of children with disabilities in Romania continues to face numerous challenges, as reported in both national and international official documents, academic research, and by non-governmental organizations. A national report presents that less than 30% of children with disabilities receive regular rehabilitation services [10].

### Bibliometric Analysis of the Topic

Bibliometric analysis is known to be a powerful research technique that provides a systematic and quantitative evaluation of scientific literature in a specific field [10]. As stated in a methodological guide bibliometric analysis is a “popular and rigorous method for exploring and analyzing large volumes of scientific data,” and it serves to “unravel the evolutionary nuances of a specific field” and “illuminate its emerging areas” [11]. The methodology is reported to offer unique advantages for the systematic review of large literature collections, mapping of research networks, analysis of interdisciplinary collaborations, and identification of future research priorities.

Bibliometric analysis in health research has become considerably more common in recent decades. Various guides and standards have been developed for its application in the health sector. Some authors reviewed 31 empirical studies and showed that the Levesque’s theoretical model has been effectively applied in healthcare access research [12]. Their results show that the model offers researchers a comprehensive perspective to evaluate the complex and dynamic process of access in both health system and community contexts. While bibliometric studies focusing specifically on children with disabilities are limited, they demonstrate that various dimensions of the field have been systematically addressed [13]. A bibliometric review on physical activity in children and adolescents with disabilities comprehensively analyzed publications from the Web of Science Core Collection between 1995 and 2023. [14] The study identified cerebral palsy, developmental coordination disorder, and autism as the most common clinical conditions in the field, also reporting that the USA and Australia are leading countries and that the dominant disciplines are neurosciences and neurology, psychology, rehabilitation, and sports sciences [14].

The impact of the COVID-19 global pandemic on healthcare access for children living with disabilities has attracted significant attention in the literature. A comprehensive review by McBride-Henry et al. (2023), which examined a total of 2201 articles from PubMed, Web of Science, CINAHL, and OVID databases between 2020 and 2023, shows that problems in healthcare access during the pandemic further deepened existing inequities [15]. Of the 81 studies analyzed, 18 specifically addressed experiences of healthcare access, while 63 examined health challenges as a secondary topic [15]. These findings bring to scholarly attention the multidimensional impact of the pandemic on individuals with disabilities.

Integrative review studies on healthcare access for individuals with intellectual and developmental disorders identify six main themes: education, information, and awareness; communication; fear and shame; participation in health decision-making processes; and the time factor [14]. All these themes seem to hinge on the need for greater care, dignity, respect, collaborative relationships, and reasonable accommodations, which brings to the fore the urgent need of a holistic approach to healthcare access for pediatric disability populations.

Our literature review verified the absence of a specialized bibliometric analysis study on healthcare access for children with disabilities, a finding that indicates a critical research gap in the field. As noted in a UNICEF report (2021a), these populations “remain largely invisible in research and in programs aimed at building more equitable, inclusive societies.” [2]. This invisibility is considered both a cause and a consequence of their exclusion.

## 2. Research Methodology

### 2.1. Research Questions

Against this backdrop, the primary research questions for our study were as follows:
What is the bibliometric profile of the global literature on access to health services for children with disabilities, and what are the publication trends?What is the structure of research in this field concerning geographical distribution, methodological approaches, and interdisciplinary collaborations?What are the core research topics, conceptual clusters, and future research directions in the field?

### 2.2. Research Aim

The main purpose of our study was to examine the international literature on access to health services for children with disabilities via bibliometric analysis and to systematically identify the research trends in the field. The objectives were to (1) determine thematic trends and conceptual evolution in the literature; (2) identify the most influential publications, authors, and institutions; (3) evaluate methodological approaches and research paradigms; (4) analyze international collaboration networks; and (5) forecast future research directions.

### 2.3. Material

The data forming the basis of the bibliometric analysis in this study were obtained from the Web of Science Core Collection (WoS) database. The scope of the search included publications listed in the Science Citation Index Expanded (SCI-Expanded), Social Sciences Citation Index (SSCI), Arts and Humanities Citation Index (AHCI), and Emerging Sources Citation Index (ESCI). A thorough keyword strategy was developed to analyze trends, collaboration networks, and knowledge production dynamics in the literature on healthcare access for children with disabilities.

The data collection process was performed on 15 June 2025, and no time restrictions were applied. The search was conducted through the “Topic” field (title, abstract, and author keywords) in the WoS database. To retrieve all relevant studies, the search string was structured as follows to encompass children with disabilities, access to health services, and the pediatric context:

“children with disabilities” OR “disabled children” OR “childhood disability” OR “pediatric disability” OR “children with special needs” OR “special needs children” OR “developmental disorders” OR “neurodevelopmental disorders” OR “intellectual disability” OR “learning disability” OR “physical disability” OR “cognitive impairment” OR “autism spectrum disorder” OR “ASD” OR “Down syndrome” OR “cerebral palsy” OR “hearing impaired children” OR “visually impaired children”

AND “health services” OR “healthcare access” OR “access to health care” OR “health service accessibility” OR “barriers to healthcare”

AND child * OR pediatric

The search query yielded a total of 1318 articles. The publications were then filtered to include only those of the “Research Article” and “Early Access” type, which resulted in a total of 1134 articles. Finally, by limiting the selection to English-language publications, we reduced the number of studies included in the analysis to 1100.

### 2.4. Methods

This study was structured following a quantitative bibliometric analysis approach. As a research method, we adopted the systematic bibliometric analysis framework [16]. In the data collection process, academic publications covering the period 1984–2025 were screened via the Web of Science Core Collection database.

This method serves the aim of the research, to systematically screen the scientific literature focused on healthcare access for children within the disability community, evaluate productivity indicators, and reveal structural patterns in the field. The analysis process was conducted on two fundamental levels: (1) performance analysis evaluated metrics such as publication counts, author productivity, and institutional and geographical distributions; (2) science mapping provided a visualization of the scientific structure through keyword co-occurrence, thematic evolution, collaboration networks, and citation relationships.

Data analysis was performed via Biblioshiny, the graphical user interface for the bibliometrix R-package, which is customized for bibliometric reviews of scientific publications and runs in R version 4.5.0. [16]. Its Bibliometrix package executes the following multidimensional analyses: performance analysis (publication counts, citation analysis, country and institutional productivity); science mapping (co-authorship, co-citation, and bibliographic coupling analyses); thematic analysis (keyword clustering, thematic mapping, and thematic evolution); and trend analysis (time series and burst detection). Through this tool, key indicators were analyzed, including the annual distribution of publications, the most cited studies, the most productive authors and institutions, international collaborations, and networks of co-citation and co-word occurrences, country-based collaboration maps, source utilization, keyword analysis, and citation patterns. The thematic structure and evolution of topics in the field were assessed with strategic mapping and thematic analysis methods.

The literature selection process was conducted transparently and systematically in accordance with the PRISMA 2020 (Preferred Reporting Items for Systematic Reviews and Meta-Analyses) guideline [17,18]. This in-depth methodology ensures a holistic evaluation of both the quantitative and qualitative dimensions of the international literature on healthcare access for pediatric disability populations. The criteria for including and excluding publications, along with the screening process, were visualized by means of a PRISMA flow diagram.

The data selection criteria are defined according to the PICOS (Population, Intervention, Comparison, Outcomes, Study design) framework used as the basis for determining inclusion and exclusion criteria [19]. This method was adapted to the bibliometric analysis format and applied as follows:Population: Academic publications concerning health services for children with developmental, intellectual, physical, or sensory disabilities (e.g., autism spectrum disorder, Down syndrome, cerebral palsy, hearing and visual impairments, learning disabilities).Intervention: Studies addressing access to health services for children with disabilities in the context of reaching healthcare, barriers within the health system, or service utilization.Comparator: As is inherent to the nature of bibliometric analysis, comparisons of productivity and collaboration levels were made between categories such as countries, institutions, authors, and publication years, rather than experimental comparisons.Outcomes: Bibliometric outputs such as publication counts, citation levels, collaboration networks, keyword patterns, thematic trends, and keyword relationships were evaluated.Study Design: The analysis included only original research articles published in the English language and categorized as “Research Article” or “Early Access.”

## 3. Results

Detailed information regarding the systematic filtering process for the publications selected in this study is visualized in the PRISMA flow diagram presented in Figure 1. Which studies were included in or excluded from the literature search can be transparently traced through this diagram. The publication evaluation and selection processes were structured according to the PICOS framework to ensure relevance to the research question. Application of this framework ensured that the studies included in the analysis were selected with a specific methodological integrity in terms of population, topic scope, comparison level, expected outcomes, and study design.

Table 1 presents the descriptive characteristics of the bibliometric dataset. The indicators shown in the table were obtained according to standard bibliometric procedures [16]. In the table, the annual growth rate is calculated according to a compound annual growth formula based on the annual distribution of publications, while the average document age is calculated as the average difference between the publication year of the articles and the last year covered by the dataset.

The average number of citations per document is calculated by dividing the total number of citations received by all documents by the total number of documents. References analyzed represent the cumulative number of cited references across the dataset. Keywords Plus (ID) refers to indexing terms generated by the database provided by Web of Science, while authors’ keywords (DE) represent words specified by the authors themselves. Total authors represent the number of contributing authors, while single-authored articles represent publications authored by only one author. The average number of co-authors per document represents the average number of authors per article and represents a measure of collaboration intensity. International co-authorships (%) indicate the extent of international scientific collaboration by expressing the proportion of documents co-authored by researchers from different countries. Finally, the distribution by document type (e.g., research article, early access) follows the WOS classification.

The final dataset created for bibliometric analysis, whose descriptive statistics are provided in Table 1, contains a total of 1100 academic publications published between 1984 and 2025. These publications were obtained from a total of 432 different sources, the majority of which were peer-reviewed journals. The annual growth rate of the documents was calculated at 8.37%. The rate shows that studies themed on healthcare access for pediatric disability populations have received increasing academic interest over time. The average publication age of the documents in the dataset stood at 8.07 years; the figure indicates that the literature in the field includes both historical and current contributions, whereas the average number of citations per publication was 25.2, a value that is significant for indicating the academic impact level of the literature.

The studies in the dataset contain a total of 34.984 different references, which reveals that the field is enriched by an interdisciplinary body of knowledge. They contain 1.712 keywords and 2.357 author-specified keywords, a finding that points to the conceptual diversity and thematic richness of the relevant literature. Among the documents contributed by a total of 5.719 different authors, only 47 are single-authored; the proportion of single-authored publications is therefore quite low. This observation shows that the field is largely shaped by multi-authored, interdisciplinary, and collaborative academic production practices. The average number of authors per document, 6.12, also supports this inference. The presence of international collaborations in 17.82% of the studies reveals that the topic attracts global scientific interest and is supported by joint research between countries. As for the distribution of document types in the dataset, 1.081 studies are categorized as “research articles” and 19 as “early access.”

The annual publication trend presented in Figure 2 illustrates the development over time in academic literature concerning access to health services for minors with disabilities. Although the first studies in this field date back to 1984, it is noteworthy that a very limited number of publications existed until the mid-2000s. However, a significant increase in the number of publications has been observed since 2015; this upward trend peaked between 2021 and 2023. Reaching the highest level in 2023 with over 90 publications shows that the topic has become a focus of international academic interest. This high level of interest can be associated with the growing importance of themes such as equity in health services, children’s rights, inclusive health policies, and the quality of life for individuals with disabilities. The data for 2025 remain partial, as the year had not concluded at the time of data collection.

The graph in Figure 3 reflects the distribution of corresponding authors by country and the breakdown of this output in terms of single-country (SCP) versus multi-country (MCP) collaborations. With 368 publications, corresponding to 33.5% of the total output, the United States stands out as by far the most productive country. It is followed by the United Kingdom (15.7%), Australia (9.5%), and Canada (9.5%), respectively. Academic interest in health services for children with disabilities is shaped by high-income countries, particularly English-speaking developed nations. Regarding multi-country publication rates, it is seen that only 8.2% (30/368) of the output from the United States, the most prolific country, is based on international collaborations. In contrast, countries such as the United Kingdom (18.5%), Canada (18.3%), and Australia (18.1%) are notable for both high productivity and more intensive international collaborations. The considerably high MCP rates in European countries such as Ireland (36.0%), Spain (36.8%), Germany (33.3%), and France (85.7%) reflect the open nature of these countries’ research collaborations.

The multi-country collaboration trends outlined above are supported by the country-based co-publication frequencies presented in Table 2. The 19 joint publications between the USA and Canada reveal the scientific integration in North America, whereas the multi-center collaborations conducted by the United Kingdom with numerous countries (Ireland, Italy, Australia, France, Spain, and Denmark) demonstrate its central role in international research networks. The eight joint publications between Australia and New Zealand also imply a strong regional academic interaction. These multifaceted international collaborations show that the field is being addressed on a global scale and is enriched by interdisciplinary approaches. 

As demonstrated by our bibliometric analysis, academic publications in the field of pediatric disability and health services are distinctly concentrated in specialized journals. (Figure 4) The data show that in autism spectrum disorders specifically, the journals *AUTISM* (n = 57) and the *Journal of Autism and Developmental Disorders* (n = 55) are the dominant publication platforms. In the literature on child health and development, *Child: Care, Health and Development* (n = 40) and *Pediatrics* (n = 30) were identified as prominent sources. In the area of intellectual disability and developmental disorders, journals such as BMJ Open (n = 22), the *Journal of Intellectual Disability Research* (n = 20), and *Research in Developmental Disabilities* (n = 20) are seen to have made significant contributions. Such findings indicate that research on this population is approached from a multidisciplinary perspective, but a more intensive accumulation of knowledge has occurred in journals specializing in autism and developmental disorders.

The Keywords Plus analysis presented in Figure 5 reveals the primary research trends and conceptual focus within the literature on children with disabilities and health services. The prominent terms in the analysis, “autism spectrum disorder” and “developmental disabilities,” show that studies in the field are concentrated on neurodevelopmental disorders. Emphases such as “mental health services” and “psychiatric disorders” indicate that the mental health needs of children within the disability community play a central role in research.

Terms such as “young children,” “health impact,” “outcomes,” “diagnosis,” and “parents” highlight the importance of early intervention and family-centered approaches. The concepts of “health services access” and “disparities” draw attention to the systemic barriers encountered, particularly in vulnerable groups.

The prominence of specific diagnostic groups like “intellectual disability” and “cerebral palsy”, alongside demographic factors such as “families” and “age,” shows that the research encompasses sociodemographic dimensions as well as clinical ones. Geographical references like “Mediterranean” and “United States” point to the existence of academic interest in regional health policies and practices.

This conceptual structure clarifies that research in the field of children experiencing disabilities and health services exhibits a multilayered structure. Studies are seen to be shaped across a wide spectrum, ranging from diagnostic classifications to mental health service access and from family dynamics to social inequalities. Our findings here are important, as they provide a conceptual framework for future research in the field.

The three-field analysis presented in Figure 6 depicts the intellectual architecture of the literature on pediatric disability populations and health services. The analysis reflects the dominant contributions of North America-based authors (Zuckerman, Brookman-Frazee, Mandell, etc.). Their work is seen to focus on themes such as autism spectrum disorder (“autism”), access issues (“access”), and family-centered interventions (“parents”).

Distribution of keywords illustrates that the clinical (“diagnosis,” “mental-health-services”) and social (“care,” “services”) dimensions of the research are intertwined. The prominence of terms like “prevalence” and “adolescents” reflects the importance given to age-specific epidemiological analyses in the field. The presence of specific diagnostic groups such as “cerebral palsy” and “developmental disabilities” establishes the heterogeneous structure of literature.

Consistency between authors’ areas of expertise and their publication preferences (e.g., autism researchers gravitating toward developmental disability journals, mental health experts toward psychiatry publications) confirms that academic production is shaped within niche areas. Such findings offer significant clues that the literature on children in this demographic and health services develops at the intersection of clinical expertise and societal needs.

An examination of the most-cited publications in the field (Table 3) highlights the prominence of studies [20,21,22]. When evaluated in terms of total citations and normalized citation scores, one work stands out distinctly from other studies with both its total citation count and average annual citations [20]. Analysis of normalized citation scores identified that relatively recent studies also exhibited high citation performance [23,24].

The thematic mapping in Figure 7 was generated by keyword plus co-word analysis using the Bibliometrix package in R. The analysis is based on research articles from the WOS between 1984 and 2025. A total of 432 articles and 1712 keywords were analyzed. The thematic map was constructed according to the method of Callon et al. (1991), which positions themes on a two-dimensional plane according to their centrality (degree of relevance) and density (degree of development) [31]. Centrality measures the degree of interaction of a theme with other themes in the network, while density assesses a theme’s internal strength and development. This scientific mapping approach consists of four theme categories: motor themes, niche themes, emerging/declining themes, and basic themes.

Themes located in the Motor Themes (Developed Themes) category, such as “autism spectrum disorders,” “adolescents,” “comorbidity,” “cerebral palsy,” “rehabilitation,” and “adolescent,” are prominent for their high centrality and high-density values. This positioning substantiates that the themes in question are both well-developed within the literature and highly interactive with other themes. The themes of “autism spectrum disorders” and “cerebral palsy” are among the mature fields that are the subject of multidisciplinary studies.

The themes in the Niche Themes category, “persons with disabilities” and “psychopathology,” are defined by high density but low centrality values. These themes represent sub-fields of study that are well-developed within specific areas of expertise but have limited interaction with other themes in the literature. The theme of “psychopathology,” in particular, is an important area of specialization that, despite its high level of development, has a limited central contribution to the overall structure of literature.

Themes in the Emerging/Declining Themes category, such as “psychiatry,” “intervention,” “anxiety,” “gender,” and “pediatrics,” have low values in terms of both density and centrality. This placement suggests that these themes are either newly entering the research area or are experiencing a decline in interest compared to previous years. Themes like “psychiatry” and “anxiety,” despite their potential in the context of mental health and service access for children with disabilities, are considered underdeveloped areas in the current literature.

The themes located in the Basic Themes category—“health services research,” “epidemiology,” “Medicaid,” “autism spectrum disorder,” “health services,” “child,” “intellectual disability,” “disabled children,” and “child health services”—possess high centrality but relatively low-density values. These themes constitute the central topics of literature, forming the methodological and theoretical foundation for the field. The central position of the “disabled children” and “child health services” themes, in particular, demonstrates that these areas form a fundamental axis in health services research.

In summary, the positioning of themes like “autism spectrum disorders” and “cerebral palsy” as strong motor themes in the upper-right quadrant of the map pinpoints the areas where the literature is concentrated. On the other hand, the fact that themes like “psychopathology” and “persons with disabilities” remain in the niche quadrant suggests that these areas, while studied in-depth, have limited interaction within the general literature. The location of themes such as “psychiatry,” “intervention,” and “gender” in the emerging/declining themes category signals that mental health services and gender-based studies are not yet sufficiently mature or are seeing diminished interest in the context of children with disabilities. The relatively low density of basic themes, such as “health services research” and “child health services,” despite their high centrality, attests to the fact that while these areas are methodologically and structurally important, they require more in-depth, content-focused research.

The results of the trend analysis presented in Figure 8 disclose that monitoring and case-focused data collection studies, such as those involving “patients,” “disabilities monitoring networks,” and “11 sites,” have shown an increase, particularly after 2015. This development signals that large-sample, multi-center, and monitoring-based research is gaining importance in the field.

A marked rise in the terms “neurodevelopmental disorders” and “autism spectrum disorder” after 2010 reflects the rapid increase in clinical and service-oriented research centered on autism and neurodevelopmental disorders. This finding testifies to the deepening of clinical interest in neurodevelopmental diagnostic groups and the corresponding health services research.

A sharp increase observed in the terms “mental-health” and “comorbidities” after 2016 corroborates that mental health problems and psychiatric comorbidities in children with disabilities are being studied more frequently in the literature. This trend suggests a need for a multidisciplinary approach in health service planning and asserts that inequities in access to mental health services have gained priority in research.

Constant use of the terms “prevalence” and “epidemiology” over time affirms that the literature rests on a strong epidemiological foundation and that prevalence data are used as a fundamental reference point in service planning for the field. The increase in the terms “care,” “access,” and “services” over the last five years evidences a research trend concentrating on access to health services, service quality, and care models.

The term “mental-retardation” began to fall out of use from the 2010s onward, being replaced by more current and inclusive terminology. This change can be evaluated as a reflection of the terminological transformation in diagnostic classifications within the academic literature. The consistent study of terms like “gross motor function” and “cerebral-palsy” since the early 2000s upholds the view that the impact of neurological disorders on physical functionality has remained a permanent fixture on the research agenda for many years. By contrast, the relatively low frequency of terms focused on patient experience and quality of life, such as “welfare,” “satisfaction,” and “performance,” points to the fact that topics like service quality and patient/parent satisfaction are still not adequately addressed in the literature, which can be construed to imply that quantitative service research in the field is open to development in terms of content richness.

Taken together, the findings indicate that in the field of pediatric disability populations and health services, there has been a research trend over the past 10–15 years concentrating on neurodevelopmental disorders, mental health, and access inequities. Terminological transformations and the increase in multi-center epidemiological studies have strengthened methodological diversification and monitoring-based data production processes in the field. However, the limited number of studies focused on patient experience, service satisfaction, and social welfare confirms that there is a literature gap in these topics, which points to potential areas for future research.

The thematic evolution analysis provided in Figure 9 visualizes how the subject headings in the literature on pediatric disability populations and health services have transformed between 1984 and 2025, and from which themes the research focus has evolved. 

Between 1984 and 2014, the primary research axes of the literature were predominantly shaped around the themes of “access,” “anxiety,” “cerebral palsy,” “children,” and “gross motor function.” During this period, clinical problems such as access inequities, the effects of neurological conditions like cerebral palsy on child health, and motor function disorders emerged as priority research areas. The inclusion of psychological factors like anxiety in previous work attests to the period’s consideration of psychosocial dimensions, albeit to a limited extent.

In the period from 2015 to 2025, a significant transformation in research focus is observed. The prominence of themes such as “care,” “diagnosis,” “disorder,” and “parents” illustrates that service delivery models, diagnostic processes, and family-centered approaches have increasingly gained priority. The appearance of “deficit-hyperactivity disorder” and “virus infection” on the research agenda can be interpreted as a reflection in the literature of health threats specific to the post-pandemic era and of neurodevelopmental disorders.

In general terms, the thematic analysis establishes that the literature on children experiencing disabilities and health services has undergone a significant transformation over approximately 40 years. It clarifies that while initially focusing on motor function disorders and access issues, the literature has now evolved into a more thorough and holistic structure that includes diagnostic processes, family support systems, and psychosocial factors. The recent prominence of research on attention deficit/hyperactivity disorder, viral infections, and parent-focused studies, in particular, reflects a broadening paradigm shift in health services, moving from the individual toward family- and community-based approaches.

The network graph provided in Figure 10 illustrates the relationships and clustering among the most frequently used keywords in the relevant publications.

The clusters, shown in different colors, represent the sub-thematic areas of the topic; they are observed to be concentrated around focal concepts such as “children,” “care,” and “prevalence.” In the network, node size represents the frequency of the word’s use, whereas the lines between nodes represent the frequency of co-occurrence. This analysis determined that themes such as access to health services, care processes, and mental health problems for children experiencing disabilities are the main areas of study highlighted in the literature. The relationships between clusters also serve as an indicator of interdisciplinary collaboration and the need for a holistic approach.

## 4. Discussion

Our study presents a thorough bibliometric analysis that systematically examines research trends concerning access to health services for pediatric disability populations. The literature in the field has been documented to have undergone a significant quantitative and qualitative development during the 1984–2025 period. The calculated annual publication growth rate of 8.37% and the distinct upward trend observed, especially after 2015, seem to confirm that the topic has become a focus of academic interest.

According to our country-based analyses, high-income countries, such as the USA, the United Kingdom, and Australia, play a dominant role in research production, which means that the perspective of developed nations largely shapes research on healthcare access for minors with disabilities. On the other hand, the intensive collaborations among European countries attest to the strength of regional research networks.

Keyword co-occurrence network analysis demonstrates that publications on pediatric disability populations and health services are shaped around specific thematic clusters. The central placement of concepts such as “children” and “care” establishes that care processes for this population and prevalence studies are among the priority topics in the field. Other clusters developing around the “autism spectrum disorder” and “health services” axis substantiate that the processes of accessing health services for children with autism spectrum disorder have gained traction in the literature. This thematic distribution constitutes a valuable resource for health policy developers and clinicians in identifying priority research areas.

Our thematic map analysis revealed a research ecosystem shaped under four distinct categories, each featuring unique patterns of scholarly engagement within the field of childhood disability healthcare. The high density and centrality values of topics such as “autism spectrum disorders” and “cerebral palsy” in the motor themes category suggest that these areas hold a mature and influential position in the literature. The predominance of studies on autism spectrum disorders underscores that neurodevelopmental disorders are a priority research topic in the context of healthcare for pediatric disability populations. The fact that topics in the niche themes category, such as “psychopathology” and “persons with disabilities,” have high density but low centrality implies that although specialized work is being performed in such areas, they have limited interaction with the broader scholarly discourse, thus suggesting intellectual fragmentation. This outcome alludes to the need for further psychopathology and disability studies equipped with a more constructive interdisciplinary approach. High centrality values of topics in the basic themes category, such as “health services research” and “child health services,” indicate that such areas form the methodological backbone of the literature. However, the relatively low density of these themes suggests a need for more content-focused research.

Thematic evolution analysis also corroborates that the literature has undergone a significant transformation over time, with research focusing on motor function disorders and access issues in earlier periods having lately evolved into a more holistic structure that includes diagnostic processes, family-centered approaches, and psychosocial factors. The recent prominence of themes like “deficit-hyperactivity disorder” and “viral infection,” in particular, demonstrates the field’s responsiveness to contemporary health challenges—a research agenda that adapts dynamically to current priorities in response to pressing public health needs.

Limitations of the study are especially linked to the analyzed databases. We have scanned only the Web of Science database, and probably other important databases have published valuable studies on this topic. Future research should also include publications from PubMed, Scopus, etc., in this type of analysis, and it should be oriented to the evolution of studies concerning the topic of disabilities at different ages and in different countries. It will also be important to emphasize this evolution in relation to national and international statistics on the topic.

## 5. Conclusions

Our study has shed light on key research trends in healthcare access for children with disabilities, identifying the current state of the field and promising directions for future research. One of the most significant contributions of the study is its emphasis on addressing this complex issue from a multidimensional perspective. The integrated execution of clinical practices, policy development, and public awareness initiatives is crucial for pediatric disability populations to benefit equally from health services. Although high-income countries currently dominate research production, expanding international collaborations and achieving more equitable resource distribution will be essential for developing truly inclusive and globally applicable solutions.

Future research is recommended to prioritize more comprehensive and detailed analyses of the social, economic, and cultural barriers that confront this population in accessing healthcare. At the intersection of caregiving, child protection systems, and social services, adopting a socio-health perspective can offer valuable insights and contributions to the field. Broader deployment of qualitative and mixed methods should afford richer insights into patient and family experiences. Importantly, cross-country differences in resources, policies, and cultural perspectives on children with disabilities can point to identifiable patterns: for example, high-income settings generally offer more structured service systems, while low- and middle-income settings emphasize the roles of extended family and community networks in accessing healthcare. Such patterns are crucial, as they can guide policymakers in designing context-sensitive strategies that leverage existing strengths and address systemic gaps. In this context, comparative insights across contexts can inform international policy dialogues and support the development of adaptable, inclusive healthcare frameworks. Developing evidence-based guidelines for policymakers and healthcare providers can help address the challenges often encountered in practice. This study has the potential to serve as a foundational reference for researchers, clinicians, and policymakers committed to advancing equitable healthcare access for children with disabilities. Additionally, as much of the existing literature focuses primarily on the perspectives of parents, guardians, and healthcare professionals, future research should prioritize studies conducted directly with children with disabilities. Incorporating children’s needs and lived experiences into research will help provide a more comprehensive and inclusive understanding of the needs and barriers they face in accessing healthcare.

## Figures and Tables

**Figure 1 healthcare-13-02106-f001:**
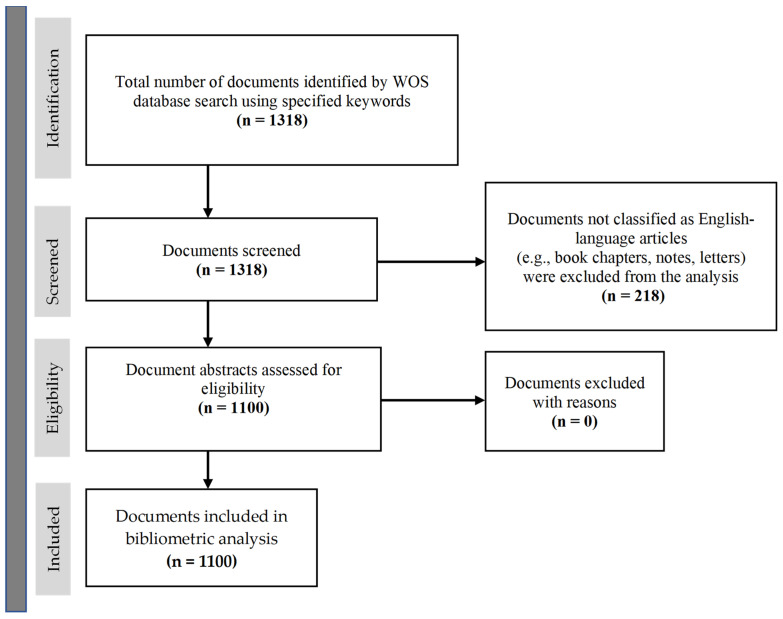
PRISMA flow chart.

**Figure 2 healthcare-13-02106-f002:**
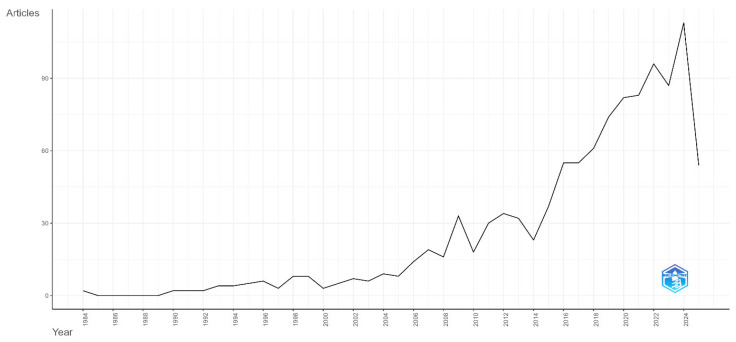
Annual scientific publication output on access to health services for children with disabilities (1984–2025).

**Figure 3 healthcare-13-02106-f003:**
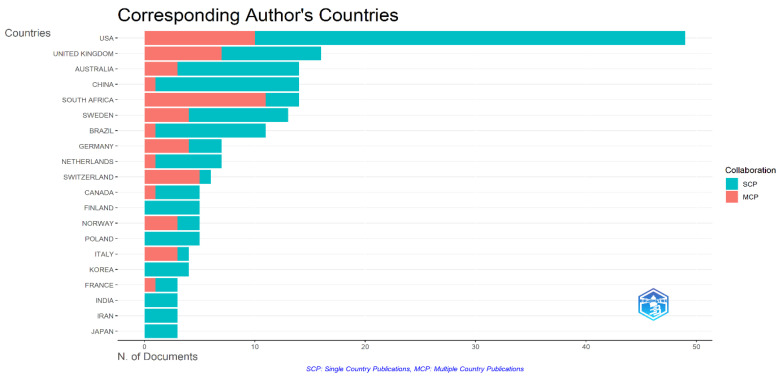
Distribution of corresponding authors by country and types of international collaboration (SCP vs. MCP).

**Figure 4 healthcare-13-02106-f004:**
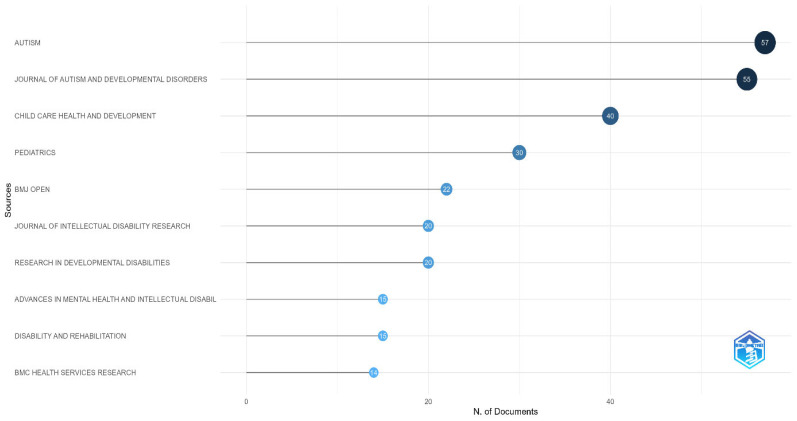
Distribution of the most prolific journals in the field of children with disabilities and health services.

**Figure 5 healthcare-13-02106-f005:**
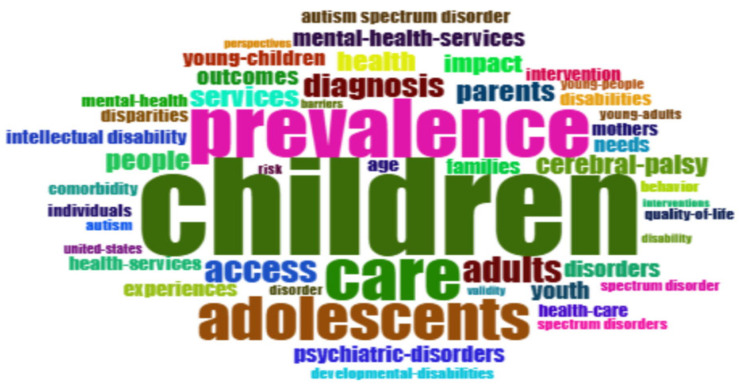
Word cloud based on keywords plus data.

**Figure 6 healthcare-13-02106-f006:**
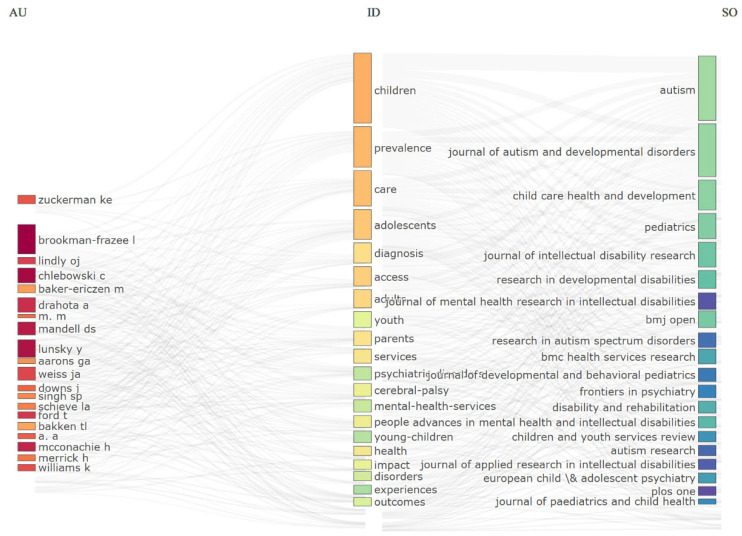
Structural dynamics of research on children with disabilities and health services: a three-fields analysis of authors, keywords, and journals.

**Figure 7 healthcare-13-02106-f007:**
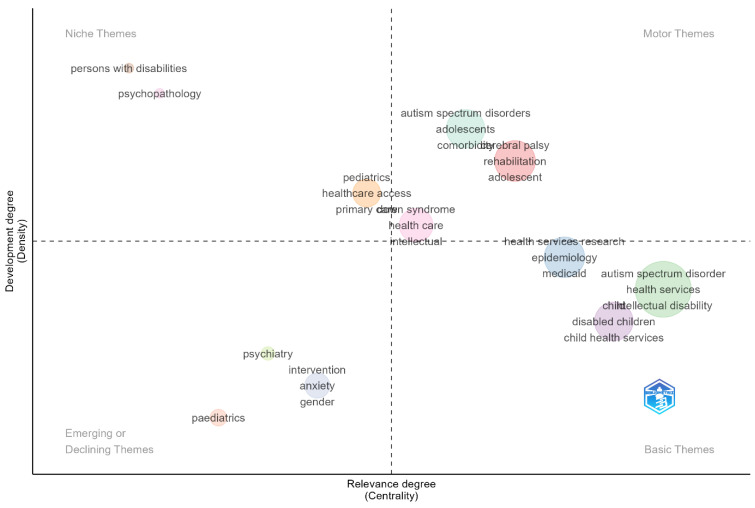
Thematic analysis.

**Figure 8 healthcare-13-02106-f008:**
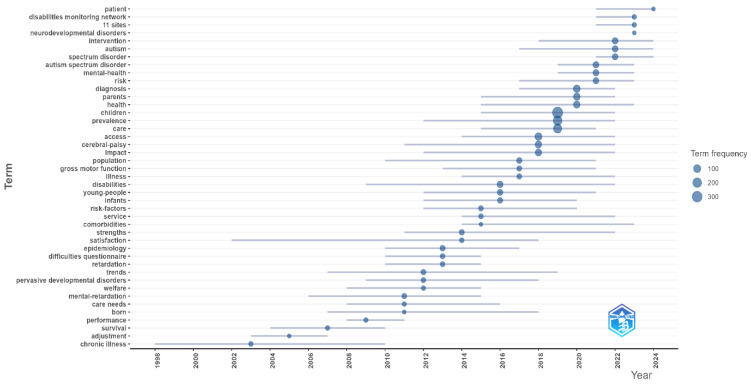
Topic trends.

**Figure 9 healthcare-13-02106-f009:**
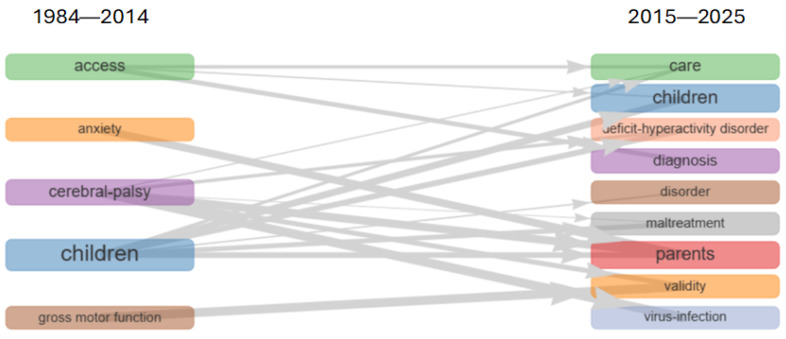
Thematic evolution analysis.

**Figure 10 healthcare-13-02106-f010:**
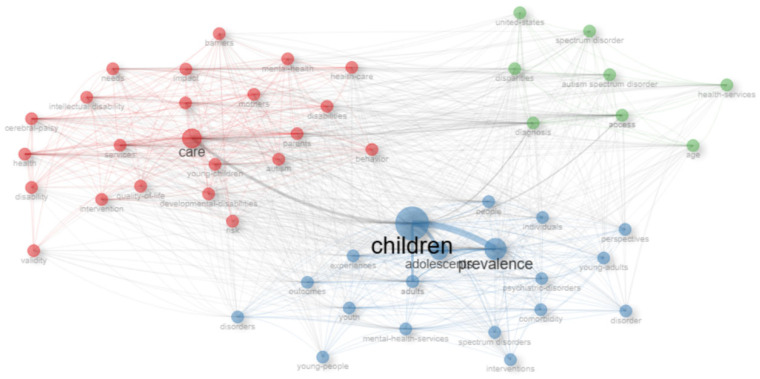
Keyword association network in publications on children with disabilities and health services.

**Table 1 healthcare-13-02106-t001:** Characteristics of the dataset used for bibliometric analysis.

Description	Value
Publication Years	1984:2025
Publication Sources	432
Documents	1100
Annual Growth Rate (%)	8.37
Average Document Age (Years)	8.07
Average Citations per Document	25.2
References Analyzed	34,984
Keywords Plus (ID) ^a^	1712
Author’s Keywords (DE) ^b^	2357
Total Authors	5719
Single-Authored Documents	46
Average Co-Authors per Document	47
International Co-Authorships (%)	6.12
Publication Years	17.82
Article	1081
Article; Early Access	19

Note. The dataset was compiled from the Web of Science Core Collection. ^a^ ID refers to “Identification” keywords indexed by the database. ^b^ DE refers to “Descriptor” keywords provided by the authors.

**Table 2 healthcare-13-02106-t002:** Frequency distribution of inter-country collaborations.

From	To	Frequency
USA	Canada	19
United Kingdom	Ireland	15
USA	United Kingdom	15
United Kingdom	Italy	13
United Kingdom	Australia	10
United Kingdom	France	10
United Kingdom	Spain	10
United Kingdom	Denmark	9
Australia	New Zealand	8
Italy	Germany	8
United Kingdom	Canada	8

**Table 3 healthcare-13-02106-t003:** The 10 most-cited articles on health services for children with disabilities.

Cited Article (From the References List)	Total Citations	Citations per Year	Normalized Citations
[20]	774	110.57	22.42
[21]	743	67.55	16.92
[22]	607	40.47	10.95
[25]	583	34.29	10.80
[26]	540	33.75	6.21
[24]	505	56.11	14.14
[27]	457	25.39	8.68
[28]	362	51.71	10.48
[29]	334	17.58	3.76
[30]	307	43.86	8.89
[23]	304	50.67	15.04

Note. Data retrieved from Web of Science on 15 June 2025. Normalized citations refer to the total citations of an article normalized by the total citations of the journal in the same year.

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
