# Peer review of "Inclusive Healthcare System for Children with Disabilities: A Bibliometric Analysis and Visualization"

_healthcare, 2025, doi:10.3390/healthcare13172106_

Round 1

Reviewer 1 Report

Comments and Suggestions for Authors

Title:

  • PLEASE express the title so that it will show what is going to be achieved by the article.

Abstract:

  • It is essential to express the objective(s) of the study explicitly in this section.
  • It is also essential to express the main question of the study explicitly in this section as well. In this case the scope of the study will be clear for the readers.
  • Please show the finding numerically. In this case, the conclusion will be sensible for the readers.

Keywords: OK.

Introduction:

  • It is good to express potential application of the results of this study in this section.

Material and method:

  • I think it is better to define the data analysis section in more detail especially in terms of “thematic analysis”, and science mapping especially for figure 7.
  • It seems essential to define items shown in table 1 in detail in this section in terms of their measurement method and interpretation (or refer to an appropriate reference).

Results:

  •  

Discussion:

  • In order to overcome with the knowledge gaps remained following presenting the results of the study, it is necessary to compare the results with other related publication. This is not evident in this section.
  • Please show the limitations of the study in this section too.

References:

Author Response

We thank the reviewer for the valuable comments, which have helped to improve the manuscript.

Comment 1: Title

Answer 1:  A new title is proposed: INCLUSIVE HEALTHCARE SYSTEM FOR CHILDREN WITH DISABILITIES: A BIBLIOMETRIC ANALYSIS AND VISUALIZATION

Comment 2: Abstract

Answer 2:  The Abstract has been revised to improve clarity and alignment with the reviewers’ suggestions. (lines 21 - 45)

Comment 3: Introduction

Answer 3:  The original contribution of the results is now highlighted in the Introduction section. (lines 94 - 102)

Comment 4:   Material and method

Answer 4:  The thematic mapping method is explained in detail before Figure 7, and a new paragraph has been introduced. (lines 503 - 512)

Comment 5: Definition of the items shown in Table 1.

Answer 5:  The descriptive statistics are now explained in detail in two paragraphs, placed before and after Table 1. (lines 33 7 -342; 349 - 360)

We thank you and hope that the revised manuscript meets the required quality standards.

Reviewer 2 Report

Comments and Suggestions for Authors

This is a excellent paper in its current form. The authors have clearly communicated the research topic, research questions and objectives. The methodology has been explained in detail making it easier for the reader to understand the step by step procedure untaken by the researchers. The results have been illustrated with clear explanations, highlighting the key findings which are then elaborated with a discussion and conclusion.

Two minor comments which I hope the authors will consider: 

  • The text in the flow chart is cut off and can be displayed more clearly. 
  • The authors mention at one point in the findings section that they have developed a conceptual framework, but then do not say much about its novelty and could possibly extend this to another sentence or two. 
  • The research highlights areas for future research, perhaps the authors would like to mention whether future research can also be conducted with children as the literature is dominated by studies with adults whether that is parents/guardians, health care professionals. Furthermore, while the study highlights most of the present research takes place in high income countries, it could say a little more about why there is a lack of research in lower-income countries when it comes to studies with children with disabilities. While financial resources is one factor, there can be others such as issues with access, safeguarding, training of researchers, complexity of disabilities which are being studied etc. If focus is given to these areas, the authors will be able to address not only the issue here but how to potentially overcome these issues for future research. 

Author Response

Thank you for your comments on the manuscript. They have contributed to its improvement.

Comment 1: The text in the flow chart - figure 1

Answer 1: Figure 1 has been revised and rearranged for improved clarity.

Comment 2: About the conceptual framework 

Answer 2: Thank you for your comment. This article proposes a conceptual framework that reflects our perspective, and it should be evaluated in this context.

Comment 3: The research highlights areas for future research, perhaps the authors would like to mention whether future research can also be conducted with children as the literature is dominated by studies with adults whether that is parents/guardians, health care professionals.

Answer 3: Thank you for this comment. A new paragraph was introduced. (lines 716-720)

We thank you and hope that the revised manuscript meets the required quality standards.

Reviewer 3 Report

Comments and Suggestions for Authors

The approach presented in the article is of interest, as it offers a thorough assessment of the scientific output in the field of childhood disabilities. It therefore constitutes a significant contribution to this area of study.
However, there are certain aspects that would benefit from further clarification:

  1. In terms of formatting: A revision is necessary (for example, the year of publication is presented inconsistently across different authors).
  2. In terms of content, there are elements that could be clarified:

a) It is stated that scientific output concerning childhood disability, particularly in relation to Attention Deficit Hyperactivity Disorder (ADHD), has experienced a “revitalisation” since 2015. The COVID-19 pandemic (2020) is also mentioned. Is there any evidence or hypothesis to explain why this shift in focus occurs specifically in 2015, beyond what has already been suggested? For instance, is there an increase in childhood illnesses in general, or is it specific to ADHD? Does this coincide with any geopolitical changes? While this may lie beyond the scope of the study, it would be a valuable aspect to highlight, given its potential influence on the shift in scholarly attention.

b) The article also refers to a multidisciplinary expansion. Have the authors observed any references to the socio-health domain in the reviewed literature? The intersection between healthcare services and caregiving (which does not necessarily fall exclusively within the healthcare sector) implicates child protection systems and social services. While acknowledging that this was not the primary focus of the analysis, could it be considered that such domains are intrinsically part of this broader approach to childhood health?

c) The article briefly addresses differences in resources, policies, and perspectives towards childhood across different contexts in the conclusion. Might these differences point to any specific patterns? Such insights would be particularly valuable from the perspective of informing public policy.

Conclusion
The article presents a compelling area of study with clear implications for public policy
and child welfare. Nevertheless, clarifying the points outlined above (a–c) would
improve the understanding and depth of the results. Revisions are therefore strongly
recommended.

Author Response

We thank the reviewer for their valuable comments, which have helped to improve the manuscript.

Comment 1: In terms of formatting

Answer 1: The format has been carefully revised following the journal’s guidelines.

Comment 2: In terms of content:

point a)  answer:  We have considered it and addressed it in the Discussion section (lines 690 - 693).

point b) answer:   We acknowledge that at the intersection of caregiving, child protection systems, and social services, adopting a socio-health perspective can offer valuable insights and contributions to the field. We add a comment on lines 706 - 707

point c) answer:  This is a valuable insight, which has been incorporated into the Conclusions section through the addition of a new paragraph (lines 707 - 714).

We sincerely thank the reviewer and hope that the revised manuscript now fulfills the journal’s quality criteria.